# Dielectric Property and Breakdown Strength Performance of Long-Chain Branched Polypropylene for Metallized Film Capacitors

**DOI:** 10.3390/ma15093071

**Published:** 2022-04-23

**Authors:** Meng Xiao, Mengdie Zhang, Haoliang Liu, Boxue Du, Yawei Qin

**Affiliations:** 1Key Laboratory of Smart Grid of Education Ministry, School of Electrical and Information Engineering, Tianjin University, Tianjin 300072, China; tjuxiaomeng@tju.edu.cn (M.X.); zmd20@tju.edu.cn (M.Z.); tju_liuhaoliang@tju.edu.cn (H.L.); 2CAS Key Laboratory of Engineering Plastics, Institute of Chemistry Chinese Academy of Sciences, Chinese Academy of Sciences, Beijing 100190, China; ywqin@iccas.ac.cn

**Keywords:** long-chain branched polypropylene, nucleating agent, crystallization, high temperature, dielectric property

## Abstract

At high temperatures, the insulation performance of polypropylene (PP) decreases, making it challenging to meet the application requirements of metallized film capacitors. In this paper, the dielectric performance of PP is improved by long-chain branching modification and adding different kinds of nucleating agents. The added nucleating agents are organic phosphate nucleating agent (NA-21), sorbitol nucleating agent (DMDBS), rare earth nucleating agent (WBG-Ⅱ) and acylamino nucleating agent (TMB-5). The results show that the long-chain branches promote heterogeneous nucleation and inhibit the motion of molecular chains, thereby enhancing the dielectric properties at high temperatures. Nucleating agents modulate the crystalline morphology of long-chain branched polypropylene (LCBPP), which leads to a decrease in the mean free path of carriers and an increase in trap energy level and trap density. Therefore, the conductivity is reduced and the breakdown strength is improved. Among the added nucleating agents, NA-21 showed a significant improvement in the electrical properties of LCBPP films. At 125 °C, compared with PP, the breakdown strength of the modified film is increased by 26.3%, and the energy density is increased by 66.1%. This method provides a reference for improving the dielectric properties of PP.

## 1. Introduction

In recent years, with the development of high-voltage direct current (HVDC) transmission, hybrid energy vehicles (HEV), oil and gas exploration, and aerospace, metallized film capacitors (MFCs) need to operate at temperatures above 100 °C [1,2,3]. Polypropylene (PP) is an important dielectric material in MFCs due to its high breakdown strength, low dielectric loss, and good self-healing properties [4]. However, the rated operating temperature of PP is only 85 °C [5]. Under the high-temperature environment, the conductivity loss of PP increases sharply, which leads to a significant decrease in the charging and discharging efficiency of the capacitors and accelerates the insulation deterioration of the films [6]. In addition, the breakdown strength and dielectric constant of the polymer affect the energy density. The sharp drop in the breakdown strength of PP with increasing temperature reduces the energy density and also increases the probability of capacitor failure [7].

The degradation of the electrical properties of PP at high temperatures is affected by its microstructure. At high temperatures, the thermal motion of the PP molecular chains is enhanced, resulting in an increase in the free volume. The breakdown strength of the films decreases due to the increase in the mean free path of the carriers [8]. Moreover, the carriers in the trap level can easily obtain energy transitions to the conduction band to become free carriers, which increases the electrical conductivity [9]. Our research found that the heat resistance of PP films can be improved by introducing long-chain branched structures on the backbone. The long-chain branches increase the frictional resistance between chains by increasing the density of entanglement points, thereby improving the thermal stability of the molecular chain [10]. Furthermore, due to the strong mobility, long-chain branches are able to pack into the lattice and promote heterogeneous nucleation [11]. The increase in spherulite density can introduce deep trap levels. Therefore, at high temperatures, LCBPP has lower conductance loss and higher breakdown strength than PP.

Controlling crystallization is an important method to improve the dielectric properties of PP films. The study found that carriers are more likely to be transported in the amorphous region due to the tighter arrangement of molecular chains in the spherulites. The addition of nucleating agents in PP can promote crystallization and limit the transport of carriers to improve the electrical properties [12]. Long-chain branches have two opposing effects on the crystallization of PP. Long-chain branching can promote heterogeneous nucleation and increase the number of spherulites [13]. In addition, the higher melt viscosity of LCBPP inhibits the growth of spherulites [11]. Therefore, it is difficult to control its crystallization only by introducing a long-chain branched structure. Our study found that the addition of an organophosphate nucleating agent (NA-21) promoted the crystallization of LCBPP to improve the dielectric properties. At present, the main applied nucleating agents in PP are α nucleating agents and β nucleating agents. The study found that the crystal form has an impact on the electrical properties of the films [14]. In addition, α nucleating agents and β nucleating agents have diverse structures and differ in their nucleating abilities, which affects the performance of films [15]. However, the current research is less concerned with the effect of different nucleating agents on the crystallization and properties of LCBPP. Therefore, this work controls the crystallization by adding small amounts of different kinds of nucleating agents and compares the effects of different nucleating agents on the dielectric properties of LCBPP.

In this paper, two α nucleating agents and two β nucleating agents were doped into LCBPP, respectively. The α nucleating agents were: NA-21 and sorbitol nucleating agent (DMDBS), and the β nucleating agents were: rare earth nucleating agent (WBG-Ⅱ) and acylamino nucleating agent (TMB-5). The effect of long-chain branches on molecular weight and its distribution as well as melt strength was investigated. Molecular structures were studied using Fourier transform infrared spectroscopy (FTIR). The crystal form, crystallization temperature, melting temperature, crystallinity and crystal morphology were investigated by X-ray diffractometer (XRD), differential scanning calorimeter (DSC) and polarized light microscope (POM), respectively. The dielectric constant and dielectric loss at room temperature as well as the conductivity and breakdown strength at different temperatures were tested. The energy storage densities of the films were calculated and compared. The results show that the addition of these nucleating agents can improve the dielectric properties of LCBPP.

## 2. Materials and Methods

2,2′-methylene-(bis-4,6-di-tert-butylphenoxy)aluminum phosphate (NA-21) and 1,3:2,4-di (3,4 dimethylbenzaldehyde sorbitol) (DMDBS) were purchased from ADEKA (Tokyo, Japan) and Milliken & Company (Spartanburg, SC, USA), respectively. Rare earth complex nucleating agent (WBG-II) was provided by Guangdong Weilinna Functional Materials Co., Ltd., Foshan, China. Substituted arylamide nucleating agent (TMB5) was from Shanxi Institute of Chemical Industry of China. The structure of isotactic PP, LCBPP, NA-21, DMDBS, WBG-II and TMB5 are shown in Figure 1.

The preparation process of the films is as follows:(1)The LCBPP was rapidly mixed with the nucleating agent in a two-roll machine at 190 °C for 10 min. The added amount of nucleating agent was 0.03 wt% and 0.05 wt%, respectively.(2)The mixture was hot-pressed at 190 °C and 20 MPa for 3 min to form a film with a thickness of 25 μm. The film was cooled to 120 °C at a pressure of 20 MPa. The naming of the samples is shown in Table 1.

The weight average molecular weight, number average molecular weight and polydispersity index of PP and LCBPP were determined by gel permeation chromatography (GPC, Waters GPC 1515, Waters Corporation, Milford, MA, USA). The sample was dissolved in 1, 2, 4-trichlorobenzene (TCB) and the solution flow rate was 1.0 mL/min. The test temperature was set to 150 °C. A Göttfert Rheotens (GÖTTFERT Werkstoff-Prüfmaschinen GmbH, Buchen, Germany) device was used to test melt strength at 200 °C. Melt strength is the maximum force measured at melt fracture. The melt flow rate was measured by the melt flow index instrument (Haake-SWO556-0031, Haake Technik GmbH, Karlsruhe, Germany) with a temperature of 230 °C and a load of 2.16 kg. FTIR spectroscopy (Thermo Scientific Nicolet iS20, Thermo Fisher Scientific, Waltham, MA, USA) was used to study groups in molecular structures. XRD (Rigaku Ultima IV, Rigaku Corporation, Tokyo, Japan) was used to study the effect of nucleating agents on crystal form. The scan rate was 5°/min. Thermal properties such as crystallization temperature, melting enthalpy and melting point were analyzed by DSC (DSC Q2000, TA Instruments, New Castle, DE, USA). The heating and cooling rate was 10 °C/min, and the test temperature was 30~250 °C. POM (DYP-90C, Shanghai Dianying Optical Instrument Co., Ltd., Shanghai, China) was used to observe the morphology of crystals. The cooling rate of the samples was 10 °C/min.

A broadband dielectric tester (Novocontrol Concept 80, Novocontrol GmbH, Frankfurt, Germany) was used to compare the dielectric constant and dissipation factor of samples at 10^−1^~10^5^ Hz at room temperature. Each sample was tested 5 times to eliminate the influence of errors, and the average value was taken as the final data. DC conductivity was measured using a three-electrode system consisting of a high-voltage electrode, a guard electrode, and a test electrode. The high-voltage electrode is a plate electrode with a diameter of 80 mm, and is connected to a DC power supply. The protective electrode is a circular ring with an inner diameter of 30 mm, which is connected to the ground. The test electrode is cylindrical with a diameter of 25 mm and is connected to an electrometer (Keithley 6517B, Keithley Instruments, Inc., Cleveland, OH, USA). The current was recorded for 30 min using the electrometer with a data sampling time of 1 s, and the electric field was kept at 20 kV/mm. The average current value of the last 3 min was regarded as the conductance current, and the DC conductivity of the films was calculated. Each sample was tested 5 times and the average value was calculated to ensure the accuracy of the test results. The test temperatures were 25 °C and 125 °C. The DC breakdown voltages of the films at 25 °C, 85 °C, 105 °C and 125 °C were tested in insulating oil using a ball-plate electrode. The ball electrode with a diameter of 10 mm is connected to the DC power supply, and the plate electrode with a diameter of 50 mm is grounded. Each sample was tested 15 times and analyzed using the Weibull distribution.

## 3. Results

### 3.1. Microstructure Characteristics

The parameters of molecular weight and distribution, melt flow rate and melt strength of PP and LCBPP are shown in Table 2, and the melt strength test results are shown in Figure 2. The ratio of weight average molecular weight (Mw) to number average molecular weight (Mn) is the polydispersity index (PDI), which is used to characterize the width of the molecular weight distribution. It can be seen that the polydispersity index of LCBPP is improved, indicating a wider molecular weight distribution due to the grafted branches on the backbone. Melt strength (MS) is Melt strength is the ability of a polymer to support its own quality in the molten state. The melt strength of LCBPP is significantly improved, which is due to the enhanced entanglement of the long-chain branches and the increased force required for molecular chain disentanglement. Compared with linear PP, the increase in entanglement point density leads to an increase in the frictional resistance between molecular chains of LCBPP and a lower melt flow rate (MFR).

The FTIR characterization is shown in Figure 3. The groups in the molecule affect the dielectric constant and loss. FTIR spectroscopy is a method of analyzing molecular structure. No new characteristic peaks appeared in LCBPP, indicating that LCBPP does not contain special groups. In addition, the wavenumbers corresponding to the characteristic peaks of FTIR did not change after adding different kinds of nucleating agents. Although NA-21, DMDBS, and TMB-5 contained functional groups, the presence of functional groups is not detected by FTIR due to the small amount of nucleating agent added.

Figure 4 shows the XRD curve. In this paper, after the introduction of long-chain branches, the 2θ values of the diffraction peaks are 14.3°, 16.9°, 18.8°, 21.3° and 22.1°, corresponding to (110), (040), (130), (111) and (131) planes, which are characteristic diffraction peaks of α crystals in PP. The angles corresponding to the diffraction peaks of the modified films with the addition of NA-21 and DMDBS did not change, and the intensities of the diffraction peaks corresponding to the (110) and (040) crystal planes increased. WBG-II and TMB5 are β nucleating agents. It can be seen that the nucleation effect of WBG-II on β crystals is relatively poor. When the additional content is 0.03 wt%, no β crystals are formed in the films. With the increase of the addition amount of WBG-II, a diffraction peak appears at 2θ = 16.2°, which corresponds to the (300) crystal plane and belongs to the characteristic diffraction peak of β crystal. Compared with WBG-II, the nucleation ability of TMB5 for β crystals is stronger for LCBPP. The intensity of the diffraction peak corresponding to the (300) crystal plane of the modified films added with TMB-5 increased more significantly.

Figure 5 shows the DSC test results, and the thermal parameters are shown in Table 3. Figure 5a is the crystallization curves of the samples. It can be seen that the crystallization temperature (*Tc*) of LCBPP is 6.3 °C higher than that of PP, due to the long-chain branches which can promote the formation of crystal nuclei. The four nucleating agents added in this paper can promote the crystallization of LCBPP, and the crystallization peaks of the modified films are shifted to the right.

Figure 5b shows the melting curves. The enhanced entanglement improves the thermal stability of the molecular chains, and thus the melting temperature (*Tm*) increases. Compared with pure LCBPP, the melting temperature of the modified films with α nucleating agent was slightly increased. In LCBPP6, LCBPP7 and LCBPP8 with β nucleating agent added, melting peaks corresponding to β crystals appeared at 152.0 °C, 152.9 °C and 153.9 °C, respectively. Due to the weak nucleation ability of WBG-II, no β crystals are formed in LCBPP5. The crystallinity (*Xc*) of the samples was calculated by melting enthalpy (*Hm*). The enhancement of entanglement inhibits the growth of spherulites, resulting in a significant decrease in the crystallinity of LCBPP. The calculation of crystallinity proves that the addition of nucleating agents promotes crystallization. There is a good match between the unit cell size of NA-21 and LCBPP, so NA-21 can induce epigenetic crystallization of molecular chains, which reduces the free energy of nucleation and improves the crystallinity [16]. Above the melting point of LCBPP, a gelled network is formed due to the hydrogen bonding of the DMBDS. The large surface of the network structure can induce heterogeneous nucleation and reduce the interfacial free energy of nucleation and crystallization [17]. The crystallinity of LCBPP4 increased by 7.3%. WBG- II is a mixed heteronuclear complex containing La^3+^ and Ca^2+^. In the process of LCBPP crystallization, La reduces the interfacial free energy of macromolecules folded perpendicular to the molecular chain direction during spherulite growth [18]. The crystallinity of LCBPP5 improves despite no β crystal formation. The formation of β crystals is induced by the binuclear complexes of Ca and La with special ligands. The nucleation principle of TMB5 can be explained by epiphytic crystallization theory. The b-axis of TMB5 is parallel to the c-axis of the β crystals of LCBPP. The distances between the periodic holes on the (b, c) crystal planes of TMB-5 and the methyl groups on the LCBPP macromolecular chains match each other. The methyl group can undergo epigenetic crystallization in the pores and cavities to form β crystals [19].

Figure 6 shows the crystalline morphology of PP, LCBPP, and modified films with α nucleating agents added. The spherulites of PP grow radially from the center of the nucleus, showing obvious α crystal characteristics. The size of the spherulites is large, and the boundaries between the spherulites are obvious. The initial nucleation density of LCBPP is higher. However, due to a large number of spherulites and high melt viscosity, the growth of spherulites is inhibited, so the size of spherulites is smaller.

The addition of NA-21 nucleating agent significantly increased the number of spherulites in LCBPP. This is because of the epiphytic crystallization of modified films, which is induced by NA-21. As the content of the nucleating agent increases, the number of crystal nuclei increases. However, due to the reduced growth space, the spherulite size is smaller. Since the melting temperature of LCBPP is lower than that of DMDBS (260 °C), DMDBS can form a gel network during the melting of LCBPP, which promotes heterogeneous nucleation. The modified films with NA-21 addition had more spherulites but lower crystallinity, which was due to the higher nucleation density hindering the growth of spherulites.

The crystal morphology of the modified films with the addition of β nucleating agents is shown in Figure 7. LCBPP6 still exhibits the morphological characteristics of α crystals, as it fails to induce the formation of β crystals. A small amount of β crystals is formed in LCBPP7 with the increase of the amount of WBG-II added. The β crystal has a lamellar structure that grows radially into divergent bundles, and the lamellae are loosely arranged. Some needle-like grains appeared during the crystallization of LCBPP7, which indicates that the addition of WBG-II led to the transformation of α crystal to β crystal. TMB-5 induced the formation of β crystals by promoting epigenetic crystallization. Crystals grow on this needle-like structure. LCBPP7 and LCBPP8 exhibit bright emission states due to the negative birefringence of the β crystal. The β crystal size is smaller, are smaller in size, with blurred edges and interpenetration. With the increase of the content of TMB-5, β crystals increased significantly.

### 3.2. Relative Permittivity and Conductivity

Figure 8 shows the relative permittivity and dissipation factor at 25 °C. After the introduction of long-chain branches, the dielectric constant of the film increases. The crystallinity of LCBPP is significantly reduced and the density of spherulites is increased, which enhances the interface polarization between the spherulites and amorphous regions. The addition of nucleating agents promotes crystallization and reduces the defects between spherulites, resulting in a decrease in polarization and dielectric constant [20]. The reason for the increased dielectric constant of LCBPP4 may be the enhanced polarity due to the -OH in DMBDS [21].

The introduction of long-chain branches inhibited the growth of spherulites and reduced the crystallinity of the films, so the dissipation factor of LCBPP was higher than that of PP. Moreover, the addition of nucleating agents makes the spherulites more compact, resulting in a decrease in relaxation loss. The increase of -OH content in LCBPP4 leads to the increase of its dielectric loss [22]. Compared with LCBPP5, the dielectric loss of LCBPP6 is higher, which may be caused by the agglomeration of WBG-II. Overall, the introduction of long-chain branches and the addition of nucleating agents have little effect on the dielectric constant and dissipation factor of the films.

The DC conductivity of PP, LCBPP and modified films with α nucleating agent added at the ambient temperature of 25 °C and 125 °C is shown in Figure 9. The standard deviation of the five measurements can be seen from the error bars. At 25 °C, the conductance loss of LCBPP is relatively high due to the decreased crystallinity. The conductivity of the films all increased significantly with increasing temperature. The conductivity of PP increased from 8.0 × 10^−14^ to 7.4 × 10^−11^ when the temperature was increased from 25 °C to 125 °C. The increase of entanglement points in LCBPP enhanced the thermal stability of molecular chains. Therefore, long-chain branches restrict the transport of carriers in the films at high temperatures. The conductance loss of LCBPP is less affected by temperature.

The conductivity loss of LCBPP can be effectively reduced by doping with α nucleating agents. The addition of α nucleating agents improves the crystallinity of LCBPP. Carriers are mainly transported in the amorphous region, so promoting crystallization can reduce conductivity loss. Compared with the DMDBS, the nucleation density of the modified film with the addition of NA-21 is higher, and more interfaces between the crystalline and amorphous regions can generate a large number of traps. Therefore, the conductivity of LCBPP4 is relatively lower. At 125 °C, the DC conductivity of LCBPP4 is 3.6 × 10^−12^ S/m, which is an order of magnitude lower than that of PP.

Figure 10 shows the DC conductivity of the modified films doped with β nucleating agent. The structure of the β crystal is loose, and there are many molecular chains at the interface of the spherulite, which leads to the increase of trap energy level. The deep traps introduced by β spherulites restrict the transport of carriers and reduce the conductance loss of the films. This work found that TMB5 has a strong ability to induce β crystals. Comparing all modified films with the addition of β nucleating agent, LCBPP8 has the lowest conductance loss. The DC conductivity of LCBPP8 was 3.9 × 10^−14^ S/m at 25 °C and 5.7 × 10^−12^ S/m at 125 °C. The DC conductivity of the modified films with WBG-II addition is relatively higher, due to the uneven dispersion. Among all modified films, LCBPP4 with the addition of NA-21 nucleating agent has the lowest conductance loss.

### 3.3. Breakdown Strength and Energy Density

Figure 11 shows the Weibull distribution of the breakdown strength of PP, LCBPP and modified films with α nucleating agents added at different temperatures. Table 4 shows the breakdown parameters of the samples. In Table 4, *Eb* is the breakdown strength of the film at a probability of 63.2%, and β is the shape parameter reflecting the dispersion of breakdown strength. The inhibition of spherulite growth by long-chain branches results in a slightly lower breakdown strength of the films at 25 °C. LCBPP readily prepares high-quality films, so its shape parameters are higher. The enhanced thermal motion of molecular chains in the amorphous region and the increase of carrier concentration leads to the decrease of the breakdown strength of the films with the increase of temperature. The entanglement of chains enhances the thermal stability of molecular chains, and the formation of a large number of spherulites affects the traps in the films. Therefore, at 125 °C, the breakdown strength of LCBPP is improved by 7.1% compared with linear PP.

The breakdown strength of the modified films with the addition of α nucleating agents is improved. The α nucleating agent promotes the crystallization of LCBPP and reduces the amorphous region of the film, which affects the transport path of carriers and improves the breakdown strength. Due to the higher spherulite density of the modified films doped with NA-21, the weak regions in the films are reduced and the breakdown strength is higher. At 25 °C, the breakdown strength of LCBPP4 is 19.7% higher than that of LCBPP. At 125 °C, the breakdown strength of LCBPP was improved by 108.0 kV/mm compared with PP. The better dispersion of DMDBS in LCBPP contributes to higher shape parameters of the modified films. As the addition amount of α nucleating agent increases, the shape parameter decreases due to the uneven dispersion.

The Weibull distribution of the breakdown strength of the modified films doped with β nucleating agent is shown in Figure 12. Although WBG-II promotes crystallization, the improvement of the breakdown strength of the modified film is not significant because only a small amount of β crystals are formed and the nucleating agents are agglomerated in the films. TMB5 promoted the formation of β crystals. The introduction of β crystals increases the trap energy level and trap density, thereby improving the breakdown strength. At 125 °C, the breakdown strength of LCBPP8 was increased by 21.4% compared with PP. Overall, the breakdown strength of the modified films with the addition of NA-21 is the highest.

Energy density refers to the energy stored in a unit volume of a dielectric and is an important parameter. The energy density of PP can be calculated by the following formula:(1)$$U_{e}=\frac{1}{2}\varepsilon_{0}\varepsilon_{r}E^{2}.$$
where *U_e_* is the energy density, *ε*_0_ and *ε_r_* are the vacuum permittivity and the relative permittivity of the dielectric material, and *E* is the electric field strength.

The theoretical maximum energy densities of the samples are shown in Figure 13. At 25 °C, the energy density of LCBPP is higher than that of PP due to the increased dielectric constant. The decrease in breakdown strength results in a decrease in energy density with increasing temperature. When the temperature is increases from 25 °C to 125 °C, the energy density of PP decreased by 38.0%. Compared with PP, the thermal stability of LCBPP molecular chains is stronger, so its energy density increases. After doping with nucleating agents, the crystallinity is increased and the crystal morphology is regulated, resulting in an increase in the breakdown strength and energy storage density. Among them, the modified films with NA-21 added have the highest energy density. At 125 °C, compared with PP, the energy density of LCBPP4 is increased by 1.11 J/cm^3^.

## 4. Discussion

The microstructure of PP has an influence on the drop in breakdown strength at high temperatures. PP is a semi-crystalline polymer. Inside the crystal region, the molecular chains are densely arranged and there are fewer defects, which limit the transport of carriers. The dielectric properties of films are mainly affected by the structure in the amorphous region [23]. In a high-temperature environment, the molecular chains of PP are prone to relative slippage, resulting in an increase in free volume [12]. The charge carriers gain energy and accelerate in the free volume, which easily leads to chain scission and a decrease in the breakdown strength of films [24]. In addition, at high temperatures, the carriers trapped by the trap center are released as free carriers under the electric field, which increases the conduction loss [9]. The microstructure of PP affects trap energy level and trap density in the films. The long-chain branched structures enhance the entanglement and hinder the slippage of molecular chains in the amorphous region. Furthermore, the introduction of long-chain branches increases the density of spherulites in the films and thus introduces deep traps. Therefore, at high temperatures, the transport of carriers is affected, and the breakdown strength of LCBPP is relatively higher.

The mechanism of the effect of crystallization on carrier transport is shown in Figure 14. The crystallinity and the spherulite density of the modified films with the addition of α nucleating agent are higher. Since the carriers are mainly transported in the space between the spherulites, the transport path is destroyed and the mean free path is shortened. In addition, a large number of spherulite interfaces can introduce deep traps [25]. Due to the more uniform distribution of spherulites in the modified film, the weak areas in the film are reduced and the local electric field in the sample is more uniform [26]. With the increase of the number of α spherulites, the conductivity loss of the films decreases and the breakdown strength increases. Therefore, the breakdown strength of the modified film with the addition of 0.05 wt% NA-21 is relatively higher. After doping with β nucleating agents, the transition from dense α crystal to loose β crystal appeared in the film. The increase in the density of spherulites inside the modified film affects the transport path of carriers. In addition, the trap energy level and trap density increase due to more molecular chains at the β spherulite interface [15]. The breakdown strength of the modified films with the addition of TMB-5 is higher due to the poor dispersion of WBG-II. Compared with the addition of TMB-5, the crystallinity of the modified films with the addition of NA-21 is higher, which limits the carrier transport. Among the four nucleating agents, NA-21 has the best effect on improving the electrical properties of LCBPP.

## 5. Conclusions

This paper proposes a method to improve the dielectric properties of PP films for MFCs by long-chain branching modification and adding different kinds of nucleating agents. The results show that the introduction of long-chain branched structures and the addition of nucleating agents can realize the co-regulation of molecular chain structure and aggregated structure to improve the breakdown strength. The conclusions can be summarized as follows:(1)The spherulite density of LCBPP increases due to the flexibility of long-chain branches. The addition of different nucleating agents promoted the crystallization of LCBPP. Among them, the unit cell size of the NA-21 matrix and LCBPP melt can be well-matched, so the initial nucleation density of the modified film is higher. TMB-5 has a good effect of inducing the formation of β crystals by promoting epigenetic crystallization.(2)Due to the decrease in crystallinity, the dielectric constant and loss of LCBPP are relatively larger. The nucleating agent induces crystallization, and the dielectric constant of the modified films decreases. By introducing long-chain branches and adding nucleating agents, the conductivity loss of the modified films is reduced and the breakdown strength and energy storage density are improved. At 125 °C, the breakdown strength of the modified films added with NA-21 is increased by 108.6 kV/mm, and the energy storage density was increased by 66.1%.(3)Long-chain branches hinder the thermal motion of chains and promote heterogeneous nucleation, thereby improving the dielectric properties at high temperatures. The addition of nucleating agents increases the nucleation density of the films and shortens the mean free path of carriers. In addition, a large number of spherulite interfaces in the modified films introduced deep traps, which affected the carrier transport.

## Figures and Tables

**Figure 1 materials-15-03071-f001:**
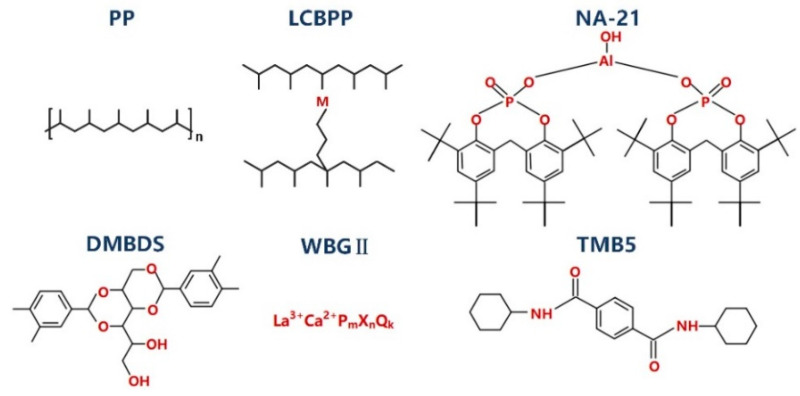
Molecular structure of PP, LCBPP and nucleating agents.

**Figure 2 materials-15-03071-f002:**
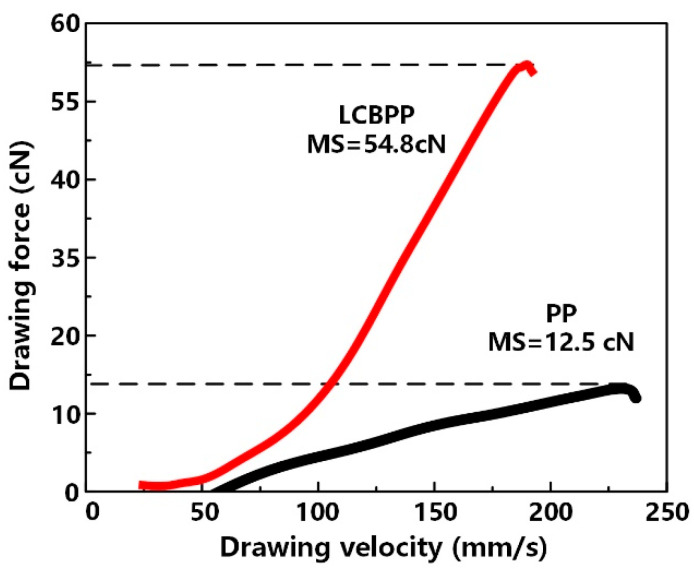
Melt strength test results for PP and LCBPP.

**Figure 3 materials-15-03071-f003:**
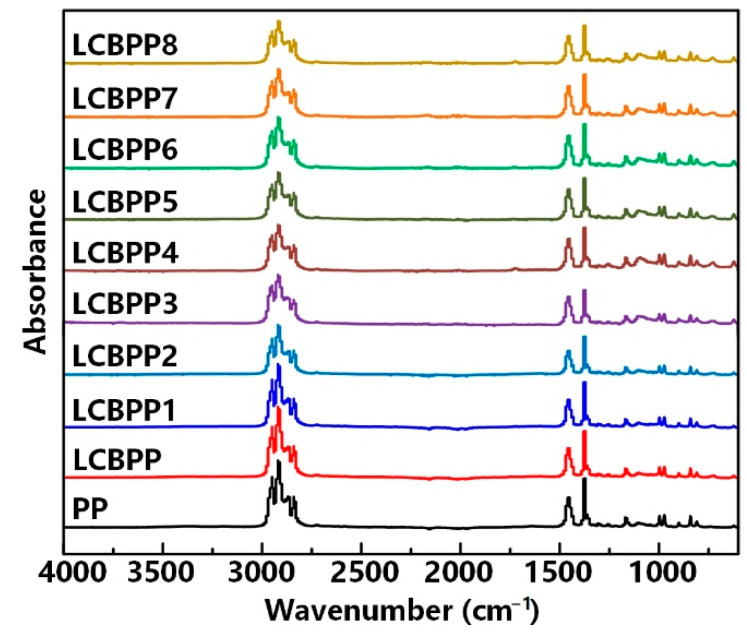
The FTIR characterization of the test samples.

**Figure 4 materials-15-03071-f004:**
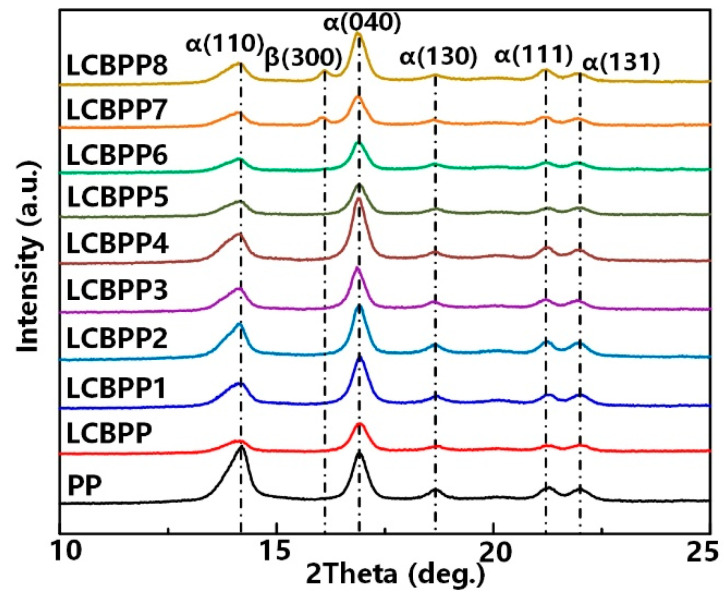
X-ray diffraction pattern of the test samples.

**Figure 5 materials-15-03071-f005:**
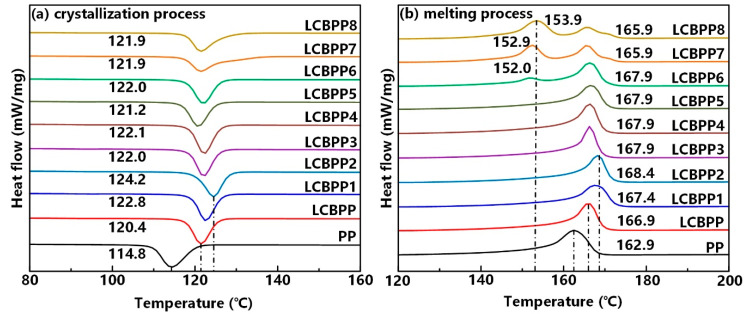
DSC results of the test samples. (**a**) crystallization process, (**b**) melting process.

**Figure 6 materials-15-03071-f006:**
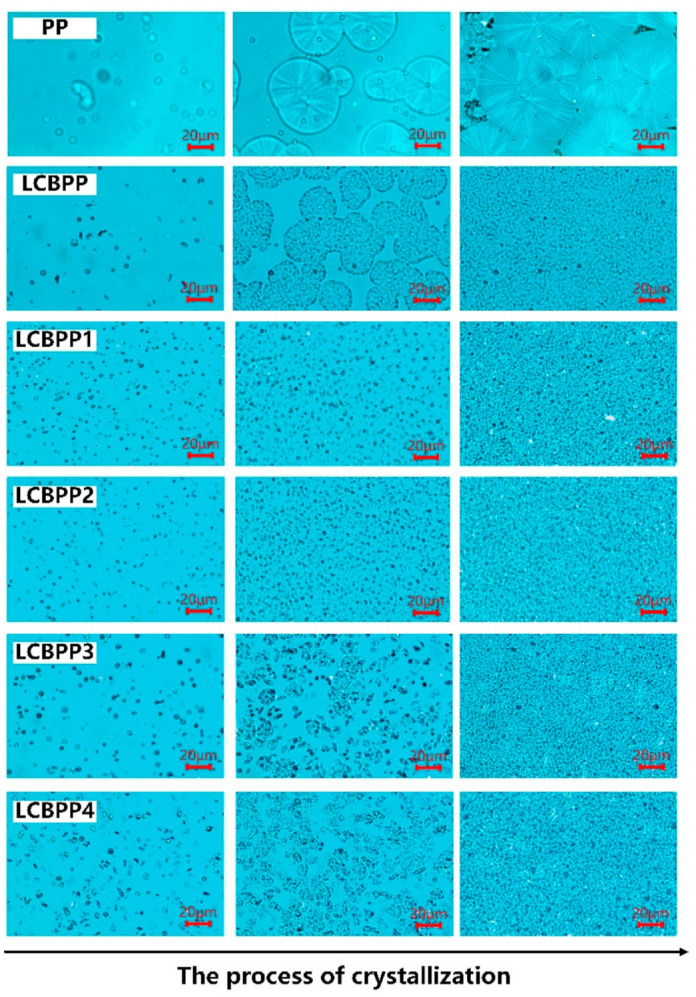
Crystalline morphology comparison of PP, LCBPP and modified films with α nucleating agents added during the process of crystallization.

**Figure 7 materials-15-03071-f007:**
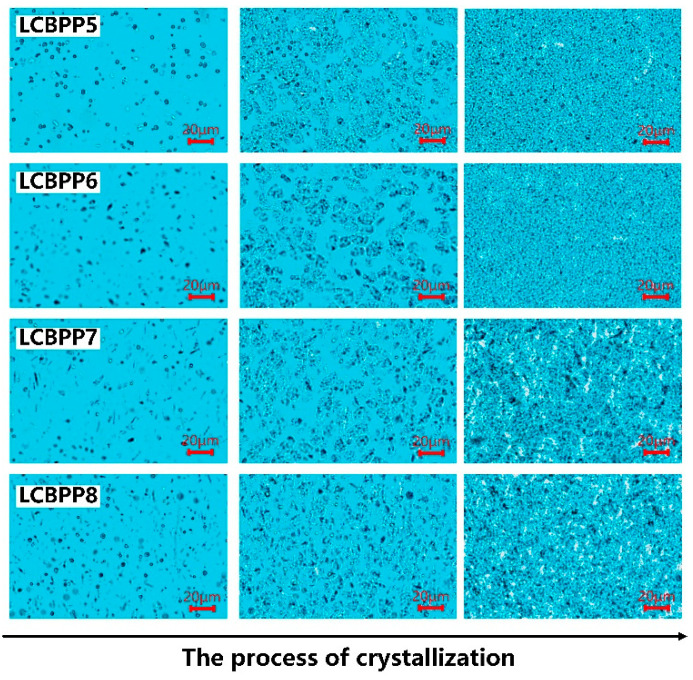
Crystalline morphology comparison of modified film with β nucleating agents added during the process of crystallization.

**Figure 8 materials-15-03071-f008:**
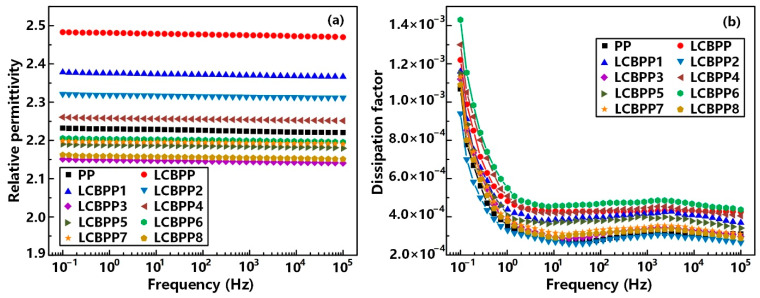
Relation between (**a**) the relative permittivity, (**b**) the dissipation factor and the frequency.

**Figure 9 materials-15-03071-f009:**
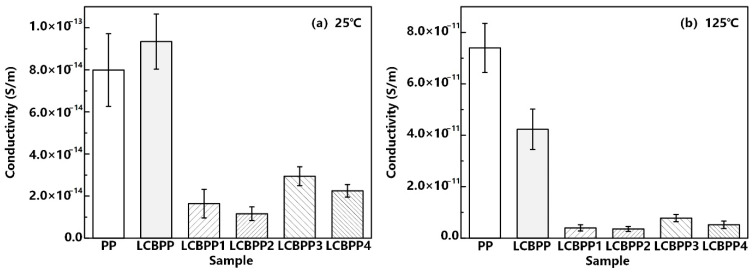
Relation between the conductivity and the α nucleating agents at different temperatures.

**Figure 10 materials-15-03071-f010:**
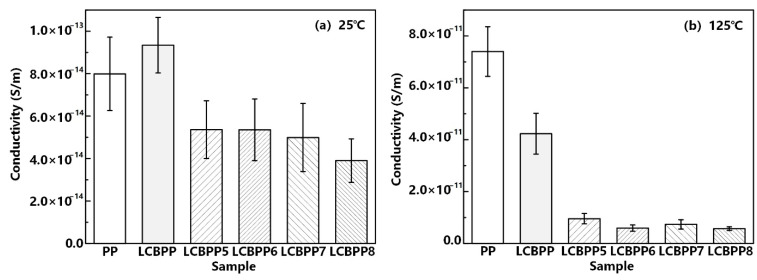
Relation between the conductivity and the β nucleating agents at different temperatures.

**Figure 11 materials-15-03071-f011:**
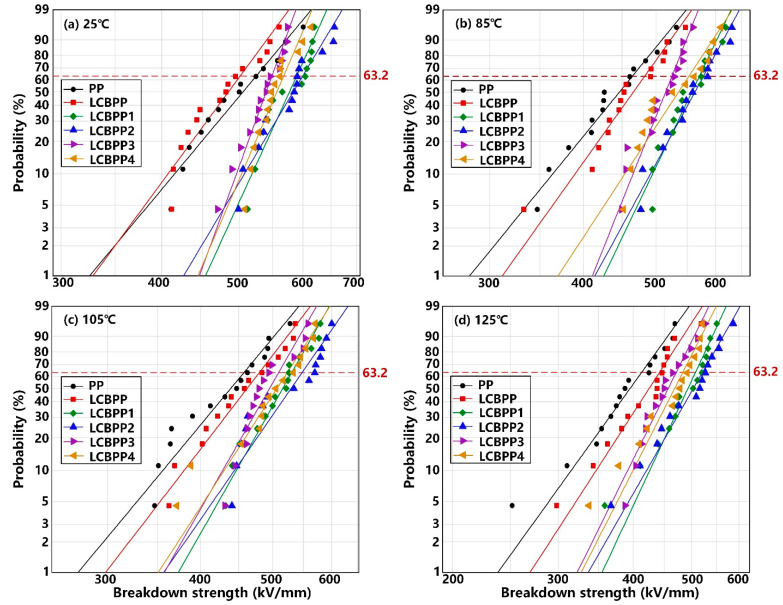
Relation between the breakdown strength and the α nucleating agents at different temperatures.

**Figure 12 materials-15-03071-f012:**
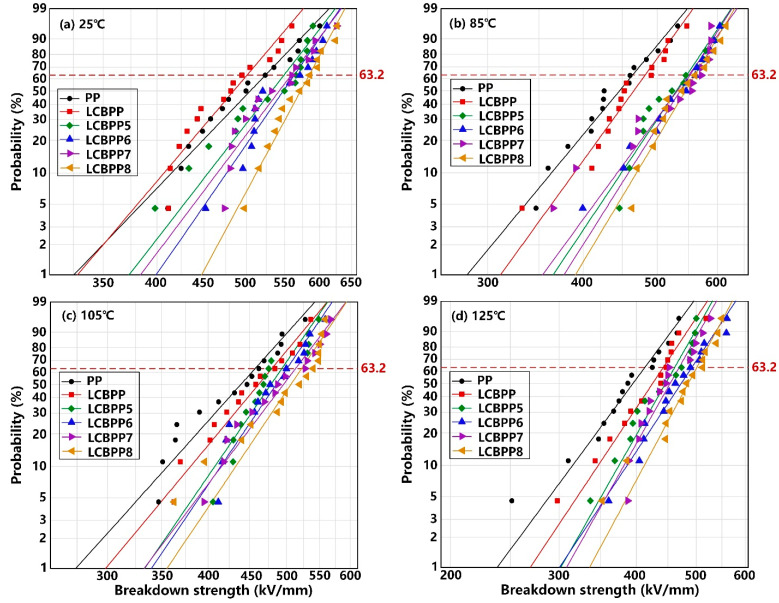
Relation between the breakdown strength and the β nucleating agents at different temperatures.

**Figure 13 materials-15-03071-f013:**
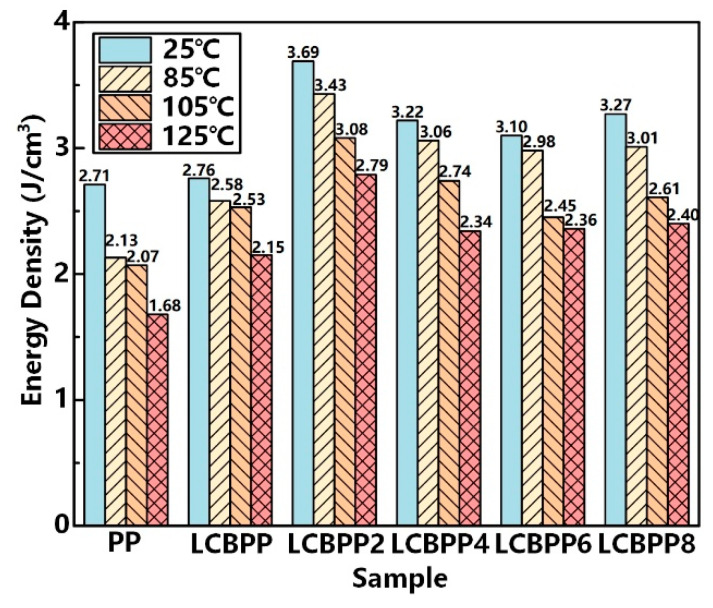
The energy density of testing samples at different temperatures.

**Figure 14 materials-15-03071-f014:**
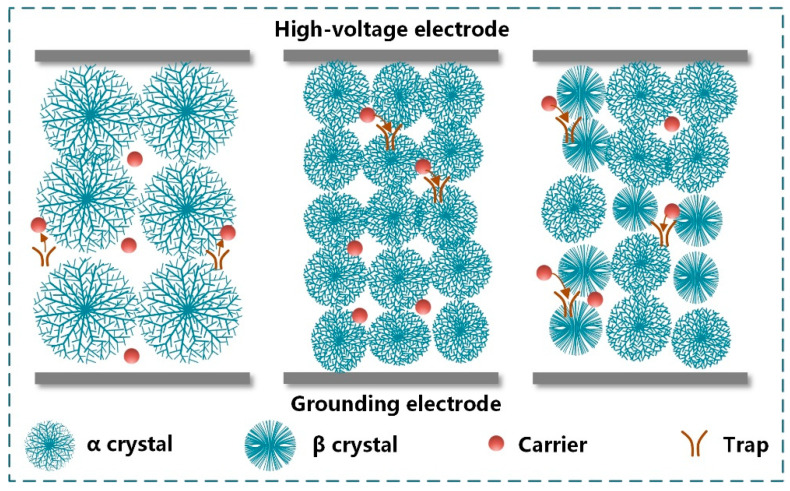
The mechanism of the effect of crystallization on carrier transport.

**Table 1 materials-15-03071-t001:** Test samples with different nucleating agents.

Sample	PP	LCBPP	LCBPP1	LCBPP2	LCBPP3	LCBPP4	LCBPP5	LCBPP6	LCBPP7	LCBPP8
Nucleating agent	-	-	NA-21	NA-21	DMDBS	DMDBS	WBG-II	WBG-II	TMB5	TMB5
Content (wt%)	0	0	0.03	0.05	0.03	0.05	0.03	0.05	0.03	0.05

**Table 2 materials-15-03071-t002:** Basic parameters of PP and LCBPP.

	Parameter	Mw (10^4^ g/mol)	Mn (10^4^ g/mol)	PDI	MS (cN)	MFR (g/10 min)
Sample						
PP		33.1	7.7	4.3	12.5	3.00
LCBPP		37.0	7.5	4.9	54.8	2.00

**Table 3 materials-15-03071-t003:** Thermal parameters of test samples.

	Sample	PP	LCBPP	LCBPP1	LCBPP2	LCBPP3	LCBPP4	LCBPP5	LCBPP6	LCBPP7	LCBPP8
Parameter											
*Tc* (°C)		114.8	120.4	122.8	124.2	122.0	122.1	121.1	122.0	121.9	121.9
*Tm* (°C)		162.9	166.9	167.4	168.4	167.9	167.9	167.9	167.9	165.9	165.9
									152.0	152.9	153.9
*Hm* (J/g)		92.6	71.2	78.8	80.9	78.7	86.5	77.0	80.8	76.1	77.3
*Xc* (%)		44.3	34.1	37.7	38.7	37.7	41.4	36.8	38.6	36.4	37.0

**Table 4 materials-15-03071-t004:** The breakdown parameters of test samples.

	Sample	PP	LCBPP	LCBPP1	LCBPP2	LCBPP3	LCBPP4	LCBPP5	LCBPP6	LCBPP7	LCBPP8
Parameter											
25 °C	*Eb* (kV/mm)	524.4	500.8	589.1	599.4	548.7	567.4	549.1	563.3	559.1	584.6
	β	9.63	10.89	17.58	13.46	22.14	18.69	11.93	13.37	12.25	17.15
85 °C	*Eb* (kV/mm)	464.3	482.3	570.7	577.8	524.6	552.5	547.0	552.4	550.0	560.6
	β	9.21	10.53	15.67	13.18	18.84	11.45	11.40	12.10	10.43	12.76
105 °C	*Eb* (kV/mm)	457.5	479.9	530.5	547.8	509.4	523.0	493.8	501.1	513.6	522.5
	β	8.97	9.68	13.13	10.76	13.01	11.55	11.72	11.82	10.62	11.95
125 °C	*Eb* (kV/mm)	412.9	442.3	507.6	521.5	471.5	483.5	461.5	492.0	462.3	501.2
	β	8.36	9.28	12.87	10.52	12.13	11.83	10.84	9.33	10.96	11.51

## Data Availability

Not applicable.

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
