# Peer review of "Dielectric Property and Breakdown Strength Performance of Long-Chain Branched Polypropylene for Metallized Film Capacitors"

_materials, 2022, doi:10.3390/ma15093071_

Round 1

Reviewer 1 Report

The paper describes dielectric properties and breakdown voltage performance of long chain branched polypropylene for metallized film capacitor. Authors say, that in high temperature, the insulation performance of polypropylene (PP) decreases. In the paper, the dielectric performance of PP was improved by long chain branching modification and adding various types of nucleating agents. Obtained results show that the long chain branches promote heterogeneous nucleation and inhibit the motion of molecular chains, thereby enhancing the dielectric properties at high temperatures. Proposed method provides a reference for improving the dielectric properties of PP.

Comments and suggestions:

  1. line 1 – I think, there should be “Breakdown”, not “Breakown”. Please correct.
  2. chapter Introduction well describes topic problems, discussed in the paper, such as film capacitor, and parameters, which have the influence on its dielectric properties. Authors present the behavior of dielectric strength of the PP and its electrical conductivity. Anyway, I did not any information about the changes of electrical permittivity, what is key factor in case of capacitors. Please complete if it is possible.
  3. Fig. 4 – please complete the unit for all used axis properties, presented on the Figures.
  4. Table 3. I think, that values in the table, could be presented also in form of charts, what would help to understand obtained results.
  5. Fig. 7 – please complete the name of the property, from “Permittivity” to “Relative electrical permittivity”.

Reviewer 2 Report

In this paper authors reported on the electrical and dielectrical performances of long-chain branched polypropylene designed for metallized Film Capacitors. In the current form the manuscript is not suitable for publication and major revision is needed as justified in the following points:

  1. The authors did not mention the instruments employed for GPC, FTIR spectroscopy, XRD, DSC, POM, dielectric spectroscopy, conductivity and breakown strength measurements. Please provide them.
  2. Nothing was mentioned about how many replicates were used in each experiment and the standard deviation of the results. These aspects are very important especially for dielectric constant, dielectric loss and conductivity parameters.
  3. For dielectric properties, the authors employed a broadband dielectric tester. However, in this work, only the dielectric constant and dielectric loss of samples at 1 KHz is illustrated. I consider that the dielectric dispersion formalism is very important for the applications like metallized film capacitors. Please provide the frequency dependences of the dielectric constant and the dielectric loss.
  4. The authors concluded that after the introduction of long-chain branches, the dielectric constant of the film increases. However, following the Figure 7, no clear trend is observed. For all considered samples, the dielectric constant varies between 2.1 and 2.3 (excepting LCBPP that exhibits a dielectric contant around 2.4). Please provide the values of dielectric constant at different frequencies.
  5. Also, the conductivity of materials is an important parameter. The numerical values of conductivity should be retrieved from dielectric measurements. Some of the relevant articles, such as Polymer 149 (2018) 73-84; Polymer 203 (2020) 122785, Advanced Functional Materials 19 (2009) 3334-3341 may be considered for evaluation of conductivity.

Round 2

Reviewer 2 Report

I agree with the revision version of the manuscript. The manuscript could be accepted in the present form.